# Diagnostic Roles of Immunohistochemical Markers CK20, CD44, AMACR, and p53 in Urothelial Carcinoma In Situ

**DOI:** 10.3390/medicina59091609

**Published:** 2023-09-06

**Authors:** Daeseon Yoo, Kyueng-Whan Min, Jung-Soo Pyo, Nae Yu Kim

**Affiliations:** 1Department of Urology, Daejeon Eulji University Hospital, Eulji University School of Medicine, Daejeon 35233, Republic of Korea; dsyoo@eulji.ac.kr; 2Department of Pathology, Uijeongbu Eulji Medical Center, Eulji University School of Medicine, Uijeongbu-si 11759, Gyeonggi-do, Republic of Korea; kyueng@eulji.ac.kr; 3Department of Internal Medicine, Uijeongbu Eulji Medical Center, Eulji University School of Medicine, Uijeongbu-si 11759, Gyeonggi-do, Republic of Korea

**Keywords:** urothelial carcinoma in situ, immunohistochemistry, meta-analysis, diagnostic test accuracy review

## Abstract

*Background and Objectives*: This study aimed to evaluate the diagnostic roles of various immunohistochemical (IHC) markers in urothelial carcinoma in situ (uCIS) through a meta-analysis and review of diagnostic test accuracy. *Materials and Methods*: The IHC markers CK20, CD44, AMACR, and p53 were evaluated in the present study. We analyzed the expression rates of the IHC markers and compared their diagnostic accuracies. *Results*: The estimated expression rates were 0.803 (95% confidence interval [CI]: 0.726–0.862), 0.142 (95% CI: 0.033–0.449), 0.824 (95% CI: 0.720–0.895), and 0.600 (95% CI: 0.510–0.683) for CK20, CD44, AMACR, and p53, respectively. In the comparison between uCIS and reactive/normal urothelium, the expression of CK20, AMACR, and p53 in uCIS was significantly higher than in reactive/normal urothelium. CD44 showed significantly lower expression in uCIS than in the reactive/normal urothelium. Among the markers, AMACR had the highest sensitivity, specificity, and diagnostic odds ratio. The AUC on SROC was the highest for CK20. *Conclusions*: In conclusion, IHC markers, such as CK20, CD44, AMACR, and p53, can be useful in differentiating uCIS from reactive/normal urothelium.

## 1. Introduction

Urothelial carcinoma in situ (uCIS) is defined as a flat proliferation of high-grade malignant cells without papillar formation. Although uCIS is most common in the urinary bladder, it can also occur throughout the urinary tract. Among urothelial malignancies, the incidence of uCIS is estimated at 1–3% [1]. The clinical implications of uCIS are well understood. uCIS is a non-muscle-invasive UC that has the potential to progress to an invasive lesion. uCIS is associated with an increased risk of recurrence [2]. In addition, the presence of uCIS indicates that there may be high-grade papillary or invasive urothelial carcinoma in the remaining tissue. It has been reported that 50–60% of patients with T1 or higher bladder cancer have CIS [1]. Bacillus Calmette–Guerin (BCG) therapy is the mainstay of uCIS treatment [3]. In this case, residual urothelial lesions or changes in the urothelium may affect the diagnosis, for example, subsequent recurrence. Differential lesions may include a reactive urothelium. Histological and cytological differentiation may be challenging; therefore, ancillary tests can be helpful. Immunohistochemical (IHC) staining, which is performed in many pathology laboratories, may be useful. Common IHC markers used in daily practice include cytokeratin CK20 and CD44. Additionally, some studies have suggested that p53 and Ki-67 might be helpful [4,5,6,7,8]. In uCIS, CK20 expression may appear to diffuse to full thickness and CD44 may be expressed in the basal layer [4,5,6,7,8,9]. However, this expression pattern is inconsistent across all cases and can present diagnostic challenges. In many cases, histology alone may not be sufficient for the diagnosis of urothelial lesions. IHC staining can also be performed to confirm tumor grade. However, individual IHC staining may not be diagnostic. Therefore, multiple IHC markers may be helpful for the diagnosis of difficult cases. In daily practice, differentiating between uCIS and reactive urothelium may not be easy using hematoxylin and eosin staining alone. In particular, a combination of positive and negative IHC markers is necessary in daily practice.

This study aimed to investigate the usefulness of IHC markers, including CK20, CD44, AMACR, and p53, in uCIS through a meta-analysis and diagnostic test accuracy (DTA) review of published articles. The expression rates of these IHC markers were estimated in uCIS and reactive/normal urothelium. In addition, a detailed comparison between uCIS and reactive/normal urothelium was performed using DTA review. The diagnostic accuracy of the DTA review was evaluated by obtaining the sensitivity, specificity, diagnostic odds ratio (OR), and area under the curve (AUC) of the summary receiver operating characteristic (SROC) curve.

## 2. Materials and Methods

### 2.1. Literature Search and Selection Criteria

Relevant articles were obtained by searching the PubMed and MEDLINE databases on 15 April 2023. The search terms used were ‘urothelial carcinoma in situ’, ‘immunohistochemistry or immunohistochemical’, and ‘CK20 or CD44 or AMACR or p53’. The titles and abstracts of all returned articles were screened for exclusion. The review articles were screened to identify additional eligible studies. English-language studies regarding CK20, CD44, AMACR, and p53 IHC expression in human uCIS were included. In addition, we included the eligible studies with information on the expression rates or diagnostic accuracy of IHC markers in uCIS or reactive/normal urothelium. Case reports and review articles were also excluded. This study was performed according to the Preferred Reporting Items for Systematic Reviews and Meta-Analyses (PRISMA) guidelines. In addition, this review has not been registered.

### 2.2. Data Extraction

Twenty-five articles were included and reviewed in this meta-analysis [4,6,7,9,10,11,12,13,14,15,16,17,18,19,20,21,22,23,24,25,26,27,28,29,30]. From eligible studies, we collected the following information: first author’s name, publication date, study location, number of patients and IHC markers analyzed, and expression rates of uCIS and reactive/normal urothelium. The expression rates of CK20, CD44, AMACR, and p53 IHC markers were collected from the eligible studies. Disagreements were resolved by consensus between two authors (Yoo D and Pyo J-S).

### 2.3. Statistical Analyses

To perform the meta-analysis, all data were analyzed using the Comprehensive Meta-Analysis software package (Biostat, Englewood, NJ, USA). The IHC expression rates of CK20, CD44, AMACR, and p53 in uCIS and reactive/normal urothelium were calculated in eligible studies. Because the eligible studies used various antibodies and evaluation criteria for various populations, a random-effects model was more suitable than a fixed-effects model. Sensitivity analysis was conducted to assess the heterogeneity of eligible studies and the impact of each study on the combined effect. Heterogeneity between studies was checked using the Q and *I*^2^ statistics and demonstrated *p*-values. To assess publication bias, Begg’s funnel plot and Egger’s test were performed. Statistical significance was set at *p* < 0.05.

In addition, a DTA review was performed using the Meta-Disc program (version 1.4) (Unit of Clinical Biostatics, Ramon y Cajal Hospital, Madrid, Spain) [31]. Pooled sensitivity and specificity were the gathered sensitivity and specificity from each eligible study, and forest plots were obtained. The summary receiver operating characteristic (SROC) curve was initially constructed by plotting the ‘sensitivity’ and ‘1-specificity’ of each study, and curve fitting was performed through linear regression using the Littenberg and Moses linear model [32]. Because heterogeneity by evaluation criteria was present, the accuracy data were pooled by fitting the SROC curve and measuring the area under the curve (AUC). An AUC close to 1 is considered a perfect test, and an AUC close to 0.5 is considered a poor test. In addition, the diagnostic OR was calculated using Meta-Disc software.

## 3. Results

### 3.1. Selection and Characteristics of Studies

A total of 107 studies were found in the database search. In the review, 115 reports were excluded due to insufficient information. The remaining reports were excluded because they studied other diseases (*n* = 20), were non-original articles (*n* = 18), used animals or cell lines (*n* = 7), or were non-English (*n* = 2). Twenty-five articles were included in this meta-analysis and DTA review (Figure 1 and Table 1).

### 3.2. Immunohistochemical Expression Rates in Urothelial Carcinoma In Situ

Firstly, a meta-analysis for the IHC expression rates was performed in uCIS. IHC expression rates of CK20, CD44, AMACR, and p53 were 0.803 (95% confidence interval [CI]: 0.726–0.862), 0.142 (95% CI: 0.033–0.449), 0.824 (95% CI: 0.720–0.895), and 0.600 (95% CI: 0.510–0.683) in uCIS, respectively (Table 2). In the sensitivity analysis, the eligible studies had no effect on the pooled estimates.

### 3.3. Comparison between Urothelial Carcinoma In Situ and Reactive/Normal Urothelium

Next, the IHC expression rates were compared between uCIS and the reactive/normal urothelium. The expressions of CK20, AMACR, and p53 were significantly higher in uCIS than in reactive/normal urothelium (Table 3; OR: 71.313, 95% CI: 30.176–168.530; OR 142.931, 95% CI: 31.109–656.697; and OR 16.774, 95% CI: 6.713–41.916, respectively; Table 3). The OR of CD44 expression between uCIS and reactive/normal urothelium was 0.016 (95% CI: 0.006–0.043).

### 3.4. Diagnostic Test Accuracy Review of Immunohistochemical Markers in Urothelial Carcinoma In Situ

The evaluated parameters of the DTA review were sensitivity, specificity, diagnostic OR, and AUC on the SROC. The pooled sensitivities of CK20, CD44, AMACR, and p53 were 0.937 (95% CI: 0.910–0.957), 0.865 (95% CI: 0.803–0.913), 0.984 (95% CI: 0.915–1.000), and 0.843 (95% CI: 0.794–0.884), respectively (Table 4). The pooled specificities of CK20, CD44, AMACR, and p53 were 0.773 (95% CI: 0.735–0.809), 0.767 (95% CI: 0.698–0.827), 0.829 (95% CI: 0.725–0.906), and 0.657 (95% CI: 0.607–0.705), respectively. The diagnostic ORs of CK20, CD44, AMACR, and p53 were 77.22 (95% CI: 30.17–172.85), 61.11 (95% CI: 23.08–161.81), 142.93 (95% CI: 31.11–656.70), and 17.17 (95% CI: 6.72–43.87), respectively. The AUC on the SROCs of CK20, CD44, AMACR, and p53 were 0.942, 0.940, 0.770, and 0.711, respectively.

## 4. Discussion

Flat urothelial lesions include uCIS and reactive urothelium [7]. Cases that are difficult to differentiate can be aided by IHC staining. In daily practice, CK20 and CD44 are useful IHC markers [11]. In uCIS, CK20 was used as a positive marker, and CD44 as a negative marker. The reactive urothelium shows the opposite IHC pattern. However, there are cases in which differentiation is difficult, even with IHC staining. The gold standard for differential diagnosis between uCIS and reactive atypia is the morphology [5]. Although IHC has limitations, the IHC panel of CK20, p53, and CD44 has potential utility. Because ureteroscopic biopsy of the upper urinary tract is difficult, the diagnostic rate is more important [33]. The usefulness of other IHC markers has been explored, as urothelial carcinoma can be triggered by genetic conditions, such as Lynch syndrome [34]. To the best of our knowledge, the present study is the first DTA review to compare IHC markers between uCIS and reactive/normal urothelium.

Urothelial carcinomas can be divided into non-muscle-invasive urothelial carcinoma (pathologic stages Ta, T1, and Tis) and muscle-invasive urothelial carcinoma (pathologic stage T2 or higher). Among non-muscle-invasive UCs, those with a flat growth pattern and no subepithelial invasion are diagnosed with uCIS. The incidence of uCIS is extremely low compared with that of papillary urothelial carcinoma [30]. There are important differences in the treatment of urothelial carcinoma based on muscle invasion. Non-muscle-invasive urothelial carcinoma, including Ta urothelial carcinoma, is diagnosed and treated by transurethral resection of bladder tumors [3]. uCIS has high-grade malignant urothelial cells. In cases with high-grade malignant urothelial cells, IHC may characteristically reveal CK20-positive and CD44-negative findings. Because some cases are CK20 negative, IHC staining may not be helpful in cases in which histological differentiation is difficult. Various IHC markers have been used in real-world diagnostics and studied for their usefulness. However, comparative studies on diagnostic accuracy are lacking and a comprehensive comparison through a DTA review would provide useful information.

In this study, we analyzed the expression of CK20, CD44, AMACR, and p53. The estimated expression rates of CK20, AMACR, and p53 were 0.803 (95% CI: 0.726–0.862), 0.824 (95% CI: 0.720–0.895), and 0.600 (95% CI: 0.510–0.683), respectively. However, CD44 had a low positive rate for uCIS (0.142, 95% CI: 0.033–0.449). We identified differences in the expression of the four markers between uCIS and reactive/normal urothelium. Of the four types of markers, the one with the critical difference between uCIS and reactive/normal urothelium was AMACR (OR, 142.931, 95% CI: 31.109–656.697). ORs of CK20 and p53 were 71.313 (95% CI: 30.176–168.530) and 16.774 (95% CI: 6.713–41.916), respectively. Furthermore, the expression comparison of uCIS with reactive/normal urothelium showed an OR of 0.016 (95% CI: 0.006–0.043). As shown in our results, CD44 was highly expressed in reactive and normal urothelium. Therefore, CD44 may be a useful negative marker for uCIS. In daily practice, CK20 and CD44 are used as combinations of positive and negative markers. The significance of this study is that additional staining of AMACR may improve the differentiation between uCIS and reactive/normal urothelium.

To evaluate the diagnostic accuracy, we performed a DTA review of the four markers. The sensitivity and specificity of the four markers were 0.843–0.984 and 0.657–0.829, respectively. Thus, these markers could be described as highly sensitive markers. However, p53 was less specific than the other markers. Based on the AUC of SROC, two markers, CK20 and CD44, had higher sensitivity and specificity values than the other AMACRs and p53. As mentioned earlier, the widely used CK20 and CD44 markers have relatively high sensitivities and specificities. In daily practice, using only positive markers may not be helpful in diagnosing CK20-negative uCIS. Therefore, it can be helpful to check for negative markers, and our results can be used as evidence for this. In the DTA review, the expression of CD44 protein was evaluated as a negative marker. The sensitivity and specificity were 0.865 (95% CI: 0.803–0.913) and 0.767 (95% CI: 0.698–0.827), respectively. The results showed a slightly lower sensitivity and similar specificity compared with those of CK20.

In this study, a DTA review of the AMACR was conducted. To our knowledge, this study is the first DTA review of the AMACR in uCIS. In the previous recommendation for IHC in bladder lesions, there was no information regarding the diagnostic role of AMACR [5]. In the present study, we evaluated AMACR expression rates in uCIS and reactive/normal urothelium. The diagnostic role of IHC markers will be useful in cases that are equivocal on hematoxylin and eosin staining in daily practice. Based on our results, it has higher sensitivity and specificity than CK20. CK20-negative uCIS was identified in up to 55.1% of cases [4,6,7,9,10,11,12,13,14,15,16,17,18,19,20,21,22,23,24,25,26,27,28,29,30]. Because we did not use a 0% threshold for evaluating CK20, there could be different distributions based on that threshold. Because some reactive/normal urothelium can show positivity in umbrella cells, there can be variations in the CK20-negative rate. However, given the large number of CK20-negative cases, it is likely that positive markers other than CK20 could be helpful in differentiating flat lesions. The thresholds used in the studies included in our meta-analysis varied slightly between studies, with some setting this as one-third of the urothelium or more, and others setting it as 5% or more. In the literature, based on one-third of the urothelium, CK20 was 100% negative in the reactive urothelium [22]. Eighty percent of uCIS cases were positive for CK20 [22]. Among the positive cases, 58.3% to two-thirds of the urothelium was positive [22]. Alston (2019) reported a 73% AMACR positivity rate in uCIS, with two-thirds of the urothelium being positive in all positive cases [10]. Aron (2013) reported that although the threshold was set at 5%, positive cases showed a diffuse and strong pattern [4]. In the present study, the sensitivity and specificity of the AMACR were 0.984 and 0.829, respectively. Based on our results, AMACR showed the highest sensitivity and specificity among the four IHC markers. However, the AUC value on SROC was the lowest among all the markers. Because three eligible studies for AMACR were included, further evaluation is needed to elucidate the diagnostic role of AMACR.

Straccia’s meta-analysis was recently published in 2022 [35]. CK20, CD44, and p53 were analyzed using the same markers as those used in our study. Unlike our study, the authors evaluated KI-67. Whereas the original meta-analysis included 15 articles, our study included 25 articles, which was a much larger number. In addition, a previous meta-analysis was performed on the expression rates of each marker. Compared with our results, they showed lower CK20 expression and higher CD44 expression. Furthermore, a comparison of expression in the reactive/normal urothelium was performed in our meta-analysis. In contrast to the previous meta-analysis, only AMACR was included in our results. In a comparison between uCIS and reactive/normal urothelium, AMACRs were highly expressed in uCIS (OR: 142.931, 95% CI: 31.109–656.697). Although the results were obtained from three papers, our results are significant. Compared with the positive marker CK20, AMACR had higher sensitivity and specificity. Although more detailed studies are needed, it is useful for CK20-negative uCIS.

This study has several limitations. First, the reactive and normal urothelium were analyzed in the same category. As there were fewer studies in both subgroups, the reactive and normal urothelium were combined. Second, the use of Ki-67 IHC staining in the evaluation of flat urothelial lesions may lead to misclassifications [23]. Third, the labeling index was excluded from this study because of its significance and confusion with positive and negative evaluations based on the baseline. Fourth, the subgroup analysis based on previous treatment or previous invasive carcinoma could not be performed because of insufficient information. Fifth, microRNAs can be biomarkers for predicting recurrence and prognosis of invasive bladder cancers [36]. The prognostic impact of these IHC markers was not evaluated. Sixth, a meta-analysis was performed using previously reported literature. Therefore, since it is excluded gray literatures, it may fundamentally have a bias.

## 5. Conclusions

In the present study, the IHC expression rates of CK20, AMACR, and p53 were significantly higher in uCIS than in reactive/normal urothelium. However, CD44 expression was significantly lower in uCIS than in reactive/normal urothelium. Based on our results, it is determined that IHC markers, such as CK20, CD44, AMACR, and p53, can be useful in differentiating uCIS from reactive/normal urothelium. AMACR, as a positive IHC marker, is useful for the identification of uCIS and differentiation from reactive/normal urothelium.

## Figures and Tables

**Figure 1 medicina-59-01609-f001:**
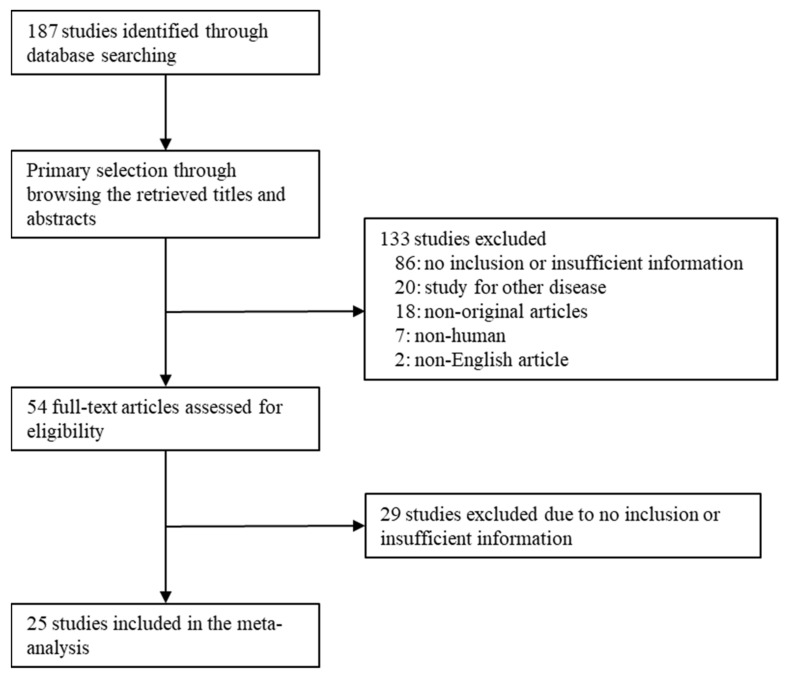
Flow chart of the searching strategy.

**Table 1 medicina-59-01609-t001:** Main characteristics of the eligible studies.

First Author	Location	Organ	No. of Patients		Interpreted Markers
			CIS	RA/Non-Neoplastic/ Normal Urothelium	
Alston 2019 [9]	USA	UB	42	30	CK20, AMACR
Arias-Stella 2018 [10]	USA	UB	69		CK20
Aron 2013 [4]	Canada	UB	43	35	CK20, CD44, AMACR
Asgari 2016 [11]	Iran	UB	20	40	CK20, CD44, p53
Barth 2018 [12]	Germany	UB	156		CK20
Dhawan 2006 [13]	UK	UB	65	56	CK20, p53
Edgecombe A [14]	Canada	UB	20	10	CK20
Garczyk 2021 [15]	Germany	UB	99		CK20
Hacek 2021 [16]	Czech Republic	UB	32		CD44
Ick 1997 [17]	USA	UB	12		p53
Jung 2014 [18]	Canada	UB	41	52	CK20, p53
Kunju 2005 [19]	USA	UB	50	50	CK20
Lombardo 2021 [20]	USA	UB	43		CK20, p53
Lopez-Beltran 2010 [21]	Spain	UB	39		CK20, p53
Mallofré 2003 [6]	USA	UB/UT	50	50	CK20, p53
McKenney 2001 [7]		UB	21	25	CK20, CD44, p53
Neal 2020 [22]	USA	UB	15	15	CK20, AMACR, p53
Nguyen 2020 [23]	USA	UB	40	40	CK20, CD44, p53
Oliva 2013 [24]	USA	UB	17	28	CK20, CD44, p53
Ozdemir 1997 [25]	Japan	UB/UT	18		p53
Sangoi 2019 [26]	USA	UB	25		CK20, CD44, p53
Sato 2011 [27]	Japan	UB	27		p53
Schmitz-Dräger 1994 [28]	Germany	UB	24		p53
Shariat 2003 [19]	USA	UB	39		p53
Tanaka 2022 [30]	Japan	UB	19		p53

CIS, carcinoma in situ; RA, reactive atypia; UB, urinary bladder; UT, urinary tract.

**Table 2 medicina-59-01609-t002:** Estimated expression rates of various immunohistochemical markers in urothelial carcinoma in situ.

	Number of Subsets	Fixed Effect [95% CI]	Heterogeneity Test [*p*-Value]	Random Effect [95% CI]	Egger’s Test [*p*-Value]
CK20	19	0.722 [0.686, 0.755]	<0.001	0.803 [0.726, 0.862]	0.002
CD44	7	0.364 [0.265, 0.476]	<0.001	0.142 [0.033, 0.449]	0.037
AMACR	3	0.824 [0.720, 0.895]	0.726	0.824 [0.720, 0.895]	0.339
p53	18	0.585 [0.537, 0.631]	<0.001	0.600 [0.510, 0.683]	0.143

CI, confidence interval.

**Table 3 medicina-59-01609-t003:** Expression ratio of various immunohistochemical markers between urothelial carcinoma in situ and reactive/normal urothelium.

	Number of Subsets	Fixed Effect [95% CI]	Heterogeneity Test [*p*-Value]	Random Effect [95% CI]	Egger’s Test [*p*-Value]
CK20	16	28.848 [17.968, 46.318]	0.001	71.313 [30.176, 168.530]	<0.001
CD44	7	0.017 [0.007, 0.043]	0.370	0.016 [0.006, 0.043]	0.110
AMACR	3	142.931 [31.109, 656.697]	0.968	142.931 [31.109, 656.697]	0.116
p53	11	8.955 [5.413, 14.814]	0.011	16.774 [6.713, 41.916]	0.008

CI, Confidence interval.

**Table 4 medicina-59-01609-t004:** Sensitivity, specificity, diagnostic odds ratio, and area under curve of summary receiver operation characteristics curve of various immunohistochemical markers in urothelial carcinoma in situ.

	Included Studies	Sensitivity (%) [95% CI]	Specificity (%) [95% CI]	Diagnostic OR [95% CI]	AUC on SROC
CK20	16	0.937 [0.910, 0.957]	0.773 [0.735, 0.809]	77.22 [30.17, 172.85]	0.942
CD44 *	7	0.865 [0.803, 0.913]	0.767 [0.698, 0.827]	61.11 [23.08, 161.81]	0.940
AMACR	3	0.984 [0.915, 1.000]	0.829 [0.725, 0.906]	142.93 [31.11, 656.70]	0.770
p53	11	0.843 [0.794, 0.884]	0.657 [0.607, 0.705]	17.17 [6.72, 43.87]	0.711

CI, confidence interval; OR, odds ratio; AUC, area under curve; SROC, summary receiver operating characteristic. * negative marker.

## Data Availability

The datasets generated during the current study are available from the corresponding author on reasonable request.

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
