# Peer review of "Diagnostic Roles of Immunohistochemical Markers CK20, CD44, AMACR, and p53 in Urothelial Carcinoma In Situ"

_medicina, 2023, doi:10.3390/medicina59091609_

Round 1
Reviewer 1 Report
The provided article discusses a meta-analysis and diagnostic test accuracy review of immunohistochemical markers (CK20, CD44, AMACR, and p53) for the differentiation of urothelial carcinoma in situ (uCIS) from reactive/normal urothelium. While the content is generally informative, there are several areas that could benefit from improvement before potential publication. A major revision is necessary to address these issues.
- Introduction:
- The introduction is concise, but it lacks proper context for readers who might not be well-versed in the subject. Consider adding more background information about uCIS and the significance of immunohistochemical markers in its diagnosis.
- The introduction could be more engaging and should clearly state the research objectives and hypotheses.
- Methods:
- Clarify the criteria used for article selection, such as the inclusion/exclusion criteria.
- Elaborate on the process of data extraction, consensus resolution, and how disagreements were addressed.
- Explain the rationale behind using a random-effects model for the meta-analysis and provide more details about the heterogeneity and sensitivity analyses.
- Results:
- The results section lacks clarity and organization. Consider breaking down this section into subsections for better readability, such as "Expression Rates of Immunohistochemical Markers," "Comparison Between uCIS and Reactive/Normal Urothelium," and "Diagnostic Test Accuracy Review."
- Include a table to present the main results, including expression rates, odds ratios, sensitivity, specificity, and AUC values.
- Discussion:
- Expand the discussion section to interpret the results more thoroughly. What do these findings mean in the context of uCIS diagnosis and treatment?
- Discuss the implications of AMACR as a useful positive marker for uCIS. How might this impact clinical practice?
- Consider addressing potential limitations and biases in the meta-analysis and discussing how they could impact the conclusions.
- Clarity and Language:
- Several sentences are convoluted and difficult to understand. Simplify the language and sentence structure to enhance readability.
- Ensure consistency in terms, abbreviations, and writing style throughout the article.
- Figures and Tables:
- Consider including figures and tables to visually present key information, such as the flowchart of study selection, forest plots for sensitivity and specificity, and summary receiver operating characteristic (SROC) curves.
- Citations and References:
- Properly format the citations and references according to the journal's guidelines.
- Include in-text citations for claims and statements, especially when referencing previous studies.
- Please include and discuss the following articles on the topic which would add value to the entire manuscript:PMID: 36964236; 35412680; 37446024.
- Conclusion:
- The conclusion is currently abrupt. Summarize the main findings, discuss their implications, and suggest potential future directions for research or clinical applications.
In summary, this article has potential, but significant revisions are needed to improve its structure, clarity, organization, and overall readability. It's important to ensure that the objectives, methods, results, and implications are clearly conveyed to the readers, making the article more informative and impactful.
Author Response
We have revised the paper to the best of our ability and added the main text.
Reviewer 1.
The provided article discusses a meta-analysis and diagnostic test accuracy review of immunohistochemical markers (CK20, CD44, AMACR, and p53) for the differentiation of urothelial carcinoma in situ (uCIS) from reactive/normal urothelium. While the content is generally informative, there are several areas that could benefit from improvement before potential publication. A major revision is necessary to address these issues.
Introduction:
The introduction is concise, but it lacks proper context for readers who might not be well-versed in the subject. Consider adding more background information about uCIS and the significance of immunohistochemical markers in its diagnosis.
Response) In many cases, histology alone may not be enough to make a diagnosis for urothelial lesion. IHC staining can also be performed to confirm tumor grade. However, individual IHC staining may be not completely diagnostic. Therefore, multiple IHC markers can be helpful in diagnosis for difficult cases. In daily practice, differentiating between uCIS and reactive urothelium may not be easy hematoxylin and eosin staining alone. Especially, a combination of positive and negative IHC markers will be necessary in daily practice. As a recommendation, we added the context in the revised manuscript.
The introduction could be more engaging and should clearly state the research objectives and hypotheses.
Response) As a recommendation, we clarified the research objectives hypotheses. In addition, we corrected the title to emphasize the objective and results.
Methods:
Clarify the criteria used for article selection, such as the inclusion/exclusion criteria.
Response) As a recommendation, we clarified the inclusion/exclusion criteria.
Elaborate on the process of data extraction, consensus resolution, and how disagreements were addressed.
Response) As a recommendation, we added the additional explanation in the revised manuscript.
Explain the rationale behind using a random-effects model for the meta-analysis and provide more details about the heterogeneity and sensitivity analyses.
Response) The eligible studies included results from different populations. When populations differ, it is appropriate to evaluate using a random-effects model. We did not apply a fixed-effects model with a heterogeneity criterion of p > 0.05. We evaluated heterogeneity and presented it in tables. Sensitivity analysis was performed and no unusual values were found. This result was added.
Results:
The results section lacks clarity and organization. Consider breaking down this section into subsections for better readability, such as "Expression Rates of Immunohistochemical Markers," "Comparison Between uCIS and Reactive/Normal Urothelium," and "Diagnostic Test Accuracy Review."
Response) As a recommendation, we divided the subsections.
Include a table to present the main results, including expression rates, odds ratios, sensitivity, specificity, and AUC values.
Response) We added the number of Table in the revised manuscript.
Discussion:
Expand the discussion section to interpret the results more thoroughly. What do these findings mean in the context of uCIS diagnosis and treatment?
Response) As a recommendation, we added the discussion for the meaning of our results.
Discuss the implications of AMACR as a useful positive marker for uCIS. How might this impact clinical practice?
Response) As a recommendation, the implications of AMACR were discussed in the revised manuscript as follow:
In the present study, we evaluated AMACR expression rates in uCIS and reactive/normal urothelium. The diagnostic role of IHC marker will be worth in cases that were equivocal on hematoxylin and eosin in daily practice.
Consider addressing potential limitations and biases in the meta-analysis and discussing how they could impact the conclusions.
Response) As a recommendation, we discussed the point in the revised manuscript as follow: A meta-analysis is performed using previously reported literature. Therefore, since it is excluded gray literatures, it may basically have bias.
Clarity and Language:
Several sentences are convoluted and difficult to understand. Simplify the language and sentence structure to enhance readability.
Response) We corrected the sentences. In addition, the certification for English editing is attached.
Ensure consistency in terms, abbreviations, and writing style throughout the article.
Response) As a recommendation, we corrected in the revised manuscript.
Figures and Tables:
Consider including figures and tables to visually present key information, such as the flowchart of study selection, forest plots for sensitivity and specificity, and summary receiver operating characteristic (SROC) curves.
Response) Of course, it would be more readable to use a figure to show key data. However, in order to express the data corresponding to one row, it is estimated that a total of 28 figures. If key findings are two or three, it might be nice to present figures. However, in order to include as many results as possible by evaluating all four IHC markers, tables were more useful than figures.
Citations and References:
Properly format the citations and references according to the journal's guidelines.
Include in-text citations for claims and statements, especially when referencing previous studies.
Please include and discuss the following articles on the topic which would add value to the entire manuscript:PMID: 36964236; 35412680; 37446024.
Response) As a recommendation, we added the comments for recommended references.
The usefulness of other IHC markers is explored, as urothelial carcinoma can be triggered by genetic conditions such as Lynch syndrome [Cerrato C, Pandolfo SD et al. 2023].
Because the ureteroscopic biopsy from upper urinary tract is not easy, the diagnostic rate is more important [Loizzo D, Pandolfo SD, 2022].
The microRNAs can be a biomarker for predicting recurrence and prognosis of invasive bladder cancers [Aveta A, Cilio S, 2023].
References
Cerrato C, Pandolfo SD, Autorino R, Panunzio A, Tafuri A, Porcaro AB, Veccia A, De Marco V, Cerruto MA, Antonelli A, Derweesh IH, Maresma MCM. Gender-specific counselling of patients with upper tract urothelial carcinoma and Lynch syndrome. World J Urol. 2023 Jul;41(7):1741-1749.
Loizzo D, Pandolfo SD, Del Giudice F, Cerrato C, Chung BI, Wu Z, Imbimbo C, Ditonno P, Derweesh I, Autorino R. Ureteroscopy and tailored treatment of upper tract urothelial cancer: recent advances and unmet needs. BJU Int. 2022 Jul;130(1):35-37.
Aveta A, Cilio S, Contieri R, Spena G, Napolitano L, Manfredi C, Franco A, Crocerossa F, Cerrato C, Ferro M, Del Giudice F, Verze P, Lasorsa F, Salonia A, Nair R, Walz J, Lucarelli G, Pandolfo SD. Urinary MicroRNAs as Biomarkers of Urological Cancers: A Systematic Review. Int J Mol Sci. 2023 Jun 29;24(13):10846.
Conclusion:
The conclusion is currently abrupt. Summarize the main findings, discuss their implications, and suggest potential future directions for research or clinical applications.
Response) As a recommendation, we corrected conclusion part in the revised manuscript.
In summary, this article has potential, but significant revisions are needed to improve its structure, clarity, organization, and overall readability. It's important to ensure that the objectives, methods, results, and implications are clearly conveyed to the readers, making the article more informative and impactful.
Response) As a recommendation, we have revised the paper to the best of our ability.

Reviewer 2 Report
Progression and recurrence of urothelial carcinoma are correlated with urothelial dysplasia and carcinoma in situ (CIS). It is sometimes challenging to distinguish CIS and dysplasia from reactive atypia based merely on histological characteristics. I congratulate the authors for their work in gathering all this data
The methodology is solid and in accordance to PRISMA guidelines required for a systematic review.
Minor issues
1. the authors must state if the article included in their systematic review included adjuvant treatments naïve pts since BCG or mitomycine can alter the urothelium and induce alterations that may be misinterpreted as dysplasia. Secondly, the type of surgery should be discussed (initial TURBT, reTURBT, random biopsies) since a previous bladder resection can induce urothelium artefacts due to coagulation
2. A short paragraph regarding the clinical utility of IHC is needed in order to broaden the audience of the manuscript to more than patologists. The International Society of Urologic Pathology consensus state that “with regard to the role of IHC in the distinction of reactive atypia from urothelial carcinoma in situ, the committee recommended that morphology remains the gold standard in this differential diagnosis” (https://pubmed.ncbi.nlm.nih.gov/25029121) The authors should discus in what clinical setting the IHC would add value to the diagnosis of CIS.
3. The article is difficult to read due to all the numbers in the results. In order to increase readability, the authors should consider whether to insert the bias analysis as a graphic rather than columns (optional requirement and of course if there is no limitation from the publisher regarding number of figures and tables)
None
Author Response
Reviewer 2.
Progression and recurrence of urothelial carcinoma are correlated with urothelial dysplasia and carcinoma in situ (CIS). It is sometimes challenging to distinguish CIS and dysplasia from reactive atypia based merely on histological characteristics. I congratulate the authors for their work in gathering all this data
The methodology is solid and in accordance to PRISMA guidelines required for a systematic review.
Response) We studied according to PRISMA guidelines. We described the comment in the manuscript. In addition, the PRISMA checklist is shown as below:
Minor issues
- the authors must state if the article included in their systematic review included adjuvant treatments naïve pts since BCG or mitomycine can alter the urothelium and induce alterations that may be misinterpreted as dysplasia. Secondly, the type of surgery should be discussed (initial TURBT, reTURBT, random biopsies) since a previous bladder resection can induce urothelium artefacts due to coagulation
Response) For example, three eligible studies included the information for AMACR. Among these articles, Alston’s and Aron’s studies included some patients with BCG treatment. In addition, in Alston’s study, some cases included the previous history of invasive carcinoma. However, the results of each study were mixed with no history of BCG or previous invasive carcinoma. The subgroup analysis based on previous treatment or previous invasive carcinoma could not be performed due to insufficient information. We included the limitation in the revised manuscript.
- A short paragraph regarding the clinical utility of IHC is needed in order to broaden the audience of the manuscript to more than patologists. The International Society of Urologic Pathology consensus state that “with regard to the role of IHC in the distinction of reactive atypia from urothelial carcinoma in situ, the committee recommended that morphology remains the gold standard in this differential diagnosis” (https://pubmed.ncbi.nlm.nih.gov/25029121) The authors should discus in what clinical setting the IHC would add value to the diagnosis of CIS.
Response) In the abstract of reference, the description is as follow:
With regard to the role of IHC in the distinction of reactive atypia from urothelial carcinoma in situ, the committee recommended that morphology remains the gold standard in this differential diagnosis and that, at best, the IHC panel of CK20/p53/CD44(s) has potential utility but is variably used and has limitations.
The gold standard in differential diagnosis between uCIS and reactive atypia is recommended the morphology. Although limitations of IHC are present, the IHC panel of CK20/p53/CD44(s) has potential utility. We analyzed the utility of AMACR as a positive marker. In addition, other IHC markers, including CK20, p53, and CD44, were analyzed. In the present study, the diagnostic roles of these IHC markers were elucidated in uCIS and reactive/normal urothelium. As a recommendation, we added the discussion in the revised manuscript.
Reference
Amin MB, Trpkov K, Lopez-Beltran A, Grignon D; Members of the ISUP Immunohistochemistry in Diagnostic Urologic Pathology Group. Best practices recommendations in the application of immunohistochemistry in the bladder lesions: report from the International Society of Urologic Pathology consensus conference. Am J Surg Pathol. 2014 Aug;38(8):e20-34.
- The article is difficult to read due to all the numbers in the results. In order to increase readability, the authors should consider whether to insert the bias analysis as a graphic rather than columns (optional requirement and of course if there is no limitation from the publisher regarding number of figures and tables)
Response) Of course, it would be more readable to use a figure to show key data. However, in order to express the data corresponding to one row, it is estimated that a total of 28 figures. If key findings are two or three, it might be nice to present figures. However, in order to include as many results as possible by evaluating all four IHC markers, tables were more useful than figures.

Round 2
Reviewer 1 Report
Considering these substantial improvements, I recommend that the paper "Diagnostic roles of immunohistochemical markers, CK20, CD44, AMACR, and p53 in urothelial carcinoma in situ " be accepted for publication in Medicina. The research presented in this manuscript contributes valuable insights to the field and aligns with the journal's scope and standards.
Thank you for your consideration, and please let me know if any further information or revisions are required.